# A Novel Earwax Method to Measure Acute and Chronic Glucose Levels

**DOI:** 10.3390/diagnostics10121069

**Published:** 2020-12-10

**Authors:** Andrés Herane-Vives, Susana Espinoza, Rodrigo Sandoval, Lorena Ortega, Luis Alameda, Allan H. Young, Danilo Arnone, Alexander Hayes, Jan Benöhr

**Affiliations:** 1Institute of Cognitive Neuroscience, Clinical, Educational & Health Psychology Department, Faculty of Brain Disease, University College London, Alexandra House, 17-19 Queen Square, Bloomsbury, London WC1N 3AZ, UK; 2Centre for Affective Disorders, Affective Disorders Research Group, Department of Psychological Medicine, Institute of Psychiatry, Psychology & Neuroscience, King’s College London, London SE5 8AF, UK; allan.young@kcl.ac.uk (A.H.Y.); danilo.arnone@uaeu.ac.ae (D.A.); alexander.hayes@kcl.ac.uk (A.H.); 3Departamento de Clínicas, Facultad de Medicina, Universidad Católica del Norte, Larrondo 1281, 1781421 Coquimbo, Chile; susana.c.espinoza@gmail.com (S.E.); rsandoval@ucn.cl (R.S.); lorena.ortega@ucn.cl (L.O.); 4Department of Psychosis Studies, Institute of Psychiatry, Psychology & Neuroscience, King’s College London, London SE5 8AF, UK; luis.alameda@kcl.ac.uk; 5Instituto de Investigación Sanitaria de Sevilla, IBiS, Hospital Universitario Virgen del Rocío, Departamento de Psiquiatría, Universidad de Sevilla, 41013 Sevilla, Spain; 6Service of General Psychiatry, Lausanne University Hospital (CHUV), 1008 Lausanne, Switzerland; 7Department of Psychiatry and Behavioural Science, College of Medicine and Health Sciences, United Arab Emirates University, 5MW2+PW Al Ain, Abu Dhabi, UAE; 8Benöhr Design Creatives, Jollystrasse 5, 81545 München, Germany; jan.benohr@gmail.com

**Keywords:** earwax, glucose, glycated hemoglobin, diabetes, diagnosis

## Abstract

Diabetes is the fourth cause of death globally. To date, there is not a practical, as well as an accurate sample for reflecting chronic glucose levels. We measured earwax glucose in 37 controls. Participants provided standard serum, glycated hemoglobin (HbA_1c_) and earwax samples at two time-points, one month apart. The specimens measured baseline fasting glucose, a follow-up postprandial glucose level and a between sample chronic glucose, calculated using the average level on the two occasions. The baseline earwax sample was obtained using a clinical method and the follow-up using a novel self-sampling earwax device. The earwax analytic time was significantly faster using the novel device, in comparison to the clinical use of the syringe. Earwax accurately reflected glucose at both assessments with stronger correlations than HbA_1c_. Follow-up postprandial concentrations were more significant than their respective fasting baseline concentrations, reflecting differences in fasting and postprandial glycemia and more efficient standardization at follow up. Earwax demonstrated to be more predictable than HbA_1c_ in reflecting systemic fasting, postprandial and long-term glucose levels, and to be less influenced by confounders. Earwax glucose measurements were approximately 60% more predictable than HbA_1c_ in reflecting glycemia over a month. The self-sampling device provided a sample that might accurately reflect chronic glycemia.

## 1. Introduction

Chronic diseases account for the largest cause of deaths globally (71%), and diabetes is the fourth among them [1]. Diabetes and other metabolic disorders are characterized by a sustained increase in glucose levels. Current measurements from “short-term” glucose specimens, such as serum, have significant limitations in the assessment of the average concentration of glucose level. This is because glucose levels can vary greatly during the day. Furthermore, day-to-day hassles during periods of stress [2], smoking [3], high blood pressure [4], Body Mass Index (BMI) [5], and physical activity [6] can affect glucose levels.

Several glucose measurements, such as fasting and postprandial levels have been standardized, aiming to provide a predictable level of glucose concentration. However, taking these tests is often demanding for patients. Furthermore, these tests still do not accurately reflect the average concentration of glucose, which is necessary to monitor the glycemic profile in metabolic disorders [7]. Indeed, these levels are usually found either below the mean, such as those seen when fasting serum glucose (FSG) samples are taken or above that average when postprandial serum glucose (PSG) ones are used [8].

Glycated hemoglobin (HbA_1c_) is a form of hemoglobin that shows a positive correlation with both FSG and PSG. Currently, HbA_1c_ is the most commonly used specimen to represent long-term glucose concentrations [9,10]. People without disease show weaker associations between HbA_1c_ and fasting or postprandial glucose levels when compared to diabetic patients [11]. This undermines the test’s ability to screen hyperglycemic level amongst the general population [2]. Fasting glucose levels show stronger associations with HbA_1c_ than postprandial glucose levels when measured in healthy people and in diabetic patients with poor glycemic control [9]. This means that HbA_1c_ could be found within a normal range in less severe diabetic patients, who frequently have dietary transgressions. This diminishes HbA_1c_ capacity to tightly monitor the mean glucose levels in glucose intolerance and mild diabetes. A more effective method for measuring the average concentration of glucose level should equally weight its postprandial and fasting levels across the day.

Currently, HbA_1c_ is used for measuring the preceding three months average plasma glucose concentration. However, this glycated protein is more greatly weighted (75%) towards plasma glucose concentrations of the previous month [12]. It is important to note that HbA_1c_ does not provide predictable information about the glycemic level over periods lesser than a month, such as those following changes (in weeks) after the prescription of hypoglycemic drugs [13,14,15]. It also has some additional limitations. Hypoglycemia for instance, is the most common adverse effect of diabetes therapy and can be associated with very severe complications [16]. However, HbA_1c_ accuracy for detecting episodes of decreased sugar levels is controversial. Whereas some studies have shown reduced HbA_1c_ associated with a larger risk of hypoglycemia [17], others have found an increased risk of hypoglycemia with higher HbA_1c_ [18,19]. Furthermore, HbA_1c_ is not a precise method, as its levels can be affected by biological variables, such as age [2] or by common illnesses, such as anemia [20] or several hemoglobinopathies (up to 7%) [21]. Even working long hours has been shown to increase HbA_1c_ levels [22]. Additionally, HbA_1c_ is an expensive and commonly unavailable lab test in some developing countries [23]. Ultimately, HbA_1c_ is an indirect approximation of the mean glucose level, given that it is the protein, not the sugar directly that is being measured. Some authors even doubt about HbA_1c_ validity as a diagnostic test for diabetes mellitus and glucose intolerance, due to its aforementioned limitations [2].

All samples, either glycemic or HbA_1c_ need to be taken from blood. This entails that sampling is expensive, since qualified workers, such as nurses are required. Blood samples may be associated with some side effects, such as bleeding, infection, and local pain. Nonetheless, the glycemic level is still the most requested lab test in primary health care centers in several countries [24,25]. The glycemia represents the third largest lab cost for some health systems [21]. HbA_1c_ is also among the most demanded lab test, and it is believed that it is still unrequested [22]. More recently, real-time continuous glucose monitoring (RT-CGM) was developed to provide readings of glucose concentration variations every five minutes for up to seven days. Nonetheless, due to its high cost this approach is not available to the population at large scale. Therefore, there is a need to find not only a more reliable specimen for measuring chronic glucose concentrations over different periods but also a harmless and more economical approach.

Few other biological samples may provide glucose concentrations. Earwax may be that specimen. Earwax is an oily secretion present within the auditory ear canal produced by apocrine and sebum glands of the ear (the ceruminous glands) [26]. Acute events mediated by the nervous system (e.g., stress reactions) are unlikely to affect the level of this secretion, since ceruminous glands are not innervated [27]. Similar to the wax produced by bees a bacteriostatic agent capable of storing sugar products (honey) in honeycombs [28,29] human wax is also capable of accumulating glucose level over long periods and may be immune to the most common strains of the epidermal flora [30,31]. Furthermore, earwax can be collected from home, without the need for specific storage or transporting conditions. We recently demonstrated that earwax reflects the chronic level of plasma concentrations of cortisol, which provides further support to the hypothesis that this specimen could also mirror peripheral chronic glucose concentrations [32].

Although glucose levels have already been measured using earwax samples elsewhere [33,34] including in diabetic patients [35], those studies did not investigate if the level found in this monosaccharide represented its long term systemic concentration. The aim of this study was to validate the use of earwax for measuring long-term glucose concentration by using a novel, self-cleaning outer-ear device which does not require specific technical expertise. We collected two right earwax samples extracted one month apart using two different methods (a conventional clinician-administered method and the earwax self-sampling device). At the same time, two glycemic samples were obtained during fasting (baseline) and after the intake of a standardized meal (follow-up) one month apart and also HbA_1c_ samples. We hypothesized that (1) the earwax self-sampling device would be an effective method to measure short and chronic glucose levels and a viable alternative to conventional methods. (2) Based on the mild/moderate associations between HbA_1c_ and glucose levels, we also expected that the novel device would be more reliable in reflecting true glycemic measures. Moreover, we predicted that (3) all follow-up concentrations would be larger than their respective baseline concentration and (4) based on the notion that ceruminous glands are not innervated, we expected that earwax, conversely to HbA_1c_ and glycemia would not be affected by short and long-term confounders mediated by the nervous system. (5) In comparison to other clinical methods, the earwax self-sampling device would reduce the time needed and cost of extraction and analysis of earwax glucose concentration (EGC).

## 2. Materials and Methods

Participants were recruited from staff and student volunteers of Universidad Católica del Norte (UCN) in Coquimbo, Chile, and from its local catchment area by public and internal advertisements. All participants were assessed by the same clinical researcher (S.E). Table 1, Table 2 and Table 3 describe the sample of thirty-seven healthy participants in detail. All participants were recruited during a Southern Hemisphere winter (between 6th of July and 3rd of August 2018). It has previously been demonstrated that different seasons vary the triglyceride composition of this secretion [36]. Asian people and people with intellectual disabilities were excluded, due to their differences in earwax characteristics [36,37]. Participants required to be free from medical illnesses (e.g., anemia, diabetes, glucose, lactose intolerance), ear pathologies (e.g., impacted earwax, perforated eardrum), and of any medication at the time of recruitment and in the previous month. Subjects were also excluded if they reported, any illicit substance use or were exposed to any severe stressor during the previous month, according to the DMS-III definition [38]. We were able to conduct a prospective case-control, rather than a prospective cross-sectional study because it has previously been found that earwax weight does not significantly differ between ear sides [32] Participants were interviewed at baseline (day = 1) and a follow-up (day = 30). During the baseline assessment a range of demographic, clinical, and environmental factors were systematically assessed (see Table 1, Table 2 and Table 3). These included the frequency and severity of the most common day-to-day environmental disturbances, using the Hassles Scale [35], and more unexpected environmental factors, such as significant life events, using the Recent Life Changes Questionnaire (RLCQ; Miller and Rahe, 1997) [39] during the month between both visits. Participants also assessed their stress perception during the last four weeks using the Perceived Stress Scale (PSS; Cohen, 1994) [40]. Anthropometric variables, such as weight, height, Body Mass Index (BMI) and waist circumference were also detailed during the final assessment. All psychometric tools were validated in Spanish versions.

At baseline, in order to collect a standardized amount of earwax secretion at the time of follow-up, the right ear of enrolled participants was cleaned using the Reiner–Alexander syringe to effectively and safely remove any earwax from outer ears [41]. It is the traditional method used by clinicians for removing impacted earwax. Participants were instructed to avoid using cotton buds or the use of any other cleaning outer-ear method during the follow-up period. During the follow-up visit, participants self-cleaned their right ears using an earwax self-sampling device, according to the manufacturer instructions (www.trears.com) and the wax collected represented the previous four weeks of earwax secretion.

Morning blood tests were obtained at both baseline and follow up visits. The baseline blood sample was obtained after 8 h of fasting whereas the follow-up sample was taken 2 h after consuming a standardized liquid meal, 236 mL of Ensure Avance^®^. FSG and HbA_1c_ levels were analyzed from baseline samples, PSG and HbA_1c_ levels were analyzed from the follow-up samples. Chronic glucose level over the preceding one-month period was calculated using the mean between the baseline and the follow-up blood sample of glycemia. Glucose concentration was extracted from earwax by using the hydrophilic fraction (see Appendix A for a detailed description of the methods). On 17th of April 2017, the local ethics committee of Universidad Católica del Norte, Coquimbo, Chile issued a resolution number 75/2017 that approved the conduction of the research. Written informed consent was obtained from all participants. Participants did not receive any financial compensation for taking part in the research.

Data were checked for normality using the Kolmogorov–Smirnov statistical test and graphics methods. All values were normally distributed (all *p* > 0.05). Therefore, we used repeated t-tests for comparing baseline and follow-up levels of all specimens. The long-term glucose concentration estimation was calculated using the mean value between FSG and PSG. Pearson correlations (R) were used to determine the association between the baseline and the follow-up EGC with their respective glycemic sample. R was also used for determining the association between the baseline and the follow-up HbA_1c_ with their respective glycemic sample. Cohen’s criteria for correlations were used: low when R = 0.1–0.3, moderate when R = 0.3–0.5 and high when R = 0.5–1.0 [42]. The coefficient of determination (R^2^) was used for comparing the predictability for measuring different glucose levels between EGC and HbA_1c_. Single linear regression analysis was used to determine the regression line of the association between different glucose samples. This statistical method was also used to determine the association between different specimens and biological and psychological variables. The level of significance was set at *p* ≤ 0.05 (two-tailed).

## 3. Results

### 3.1. Socio-Demographic, Anthropometric, and Psychological Variables Results

The sample consisted of 37 young healthy individuals (mean age 29.9 years), 54.1% women of normal weight, BMI, and waist circumference with little exposure to severe hassles or life events (see Table 1, Table 2 and Table 3 for details).

### 3.2. Baseline and Follow-Up Samples Comparisons

Follow-up postprandial concentrations using all specimens, e.g., EGC, HbA_1c_, and glycemia were significantly larger than their respective fasting baseline concentrations (see Table 4 for details).

### 3.3. Time Needed to Analyse Earwax Glucose Concentration vs. Blood Based Estimations

The self-sampling device earwax extraction time was considerably faster (04:37 h) vs. Reiner–Alexander syringe (12:20 h) (see Table 5 for details).

### 3.4. Estimated Costs Related to Different Sampling Methods for Measuring Chronic Glucose over a Three-Month Period

While earwax glucose and HbA_1c_ analytic costs were similar, we found that RT-CGM is a significantly more expensive method for measuring chronic glucose levels (see Table 6 for details).

### 3.5. Correlation of Earwax Glucose Concentration (EGC) and Glycated Hemoglobin (HbA_1c_) with Glycemic Levels

Earwax glucose concentration strongly positively correlated with all glycemic measurements (all R ≥ 0.62, R^2^ ≥ 38; *p* < 0.01). HbA_1c_ associations with glycemic levels exhibited low to moderate correlations across all the measurements (all R ≤ 0.55, R^2^ ≤ 0.30 and 0.10 < *p* < 0.01) (Table 7 and Figure 1). The strongest observed HbA_1c_ association was with the mean glycemic level at baseline (R = 0.55, R^2^ = 0.30, *p* < 0.001) and the lowest between follow up HbA_1c_ and mean blood sugar (R = 0.35, R^2^ = 0.12 *p* = 0.03) (Table 7 and Figure 1iii.b,iv.b). The lowest correlation between EGC and glycemic levels was at baseline with the mean blood sugar (R = 0.62, R^2^ = 0.38, *p* < 0.01) and the strongest at follow-up with PSG (R = 0.90, R^2^ = 0.81 *p* < 0.001) (see Table 7 and Figure 1ii.a, iii.a).

### 3.6. Accuracy of Earwax Glucose Concentration (EGC) and Glycated Hemoglobin (HbA_1c_) in Measuring Chronic Glucose Concentration

EGC was 59% more accurate in predicting glucose levels than HbA_1c_ for measuring longitudinal (chronic) glucose concentration over the two time points (Follow-up-EGC/Mean glucose level correlation: R = 0.84, R^2^ = 0.71; Follow-up- HbA_1c_/Mean glucose level correlation R = 0.35, R^2^ = 0.12) (see Table 7 and Figure 1iv.a,b).

### 3.7. Effect of Covariates on Earwax Glucose Concentration (EGC) and Glycated Hemoglobin (HbA_1c_) Correlation with Glycemic Levels

Earwax samples were not affected by any of the covariates considered (all *p* > 0.05). HbA_1c_ levels were affected by age at follow-up (*p* < 0.01) and tobacco use was negatively associated with FSG (*p* = 0.01) and PSG levels (*p* = 0.02). Increasing level of education were associated with increased HbA_1c_ and PSG levels at follow-up (both *p* < 0.05) (Table 8).

## 4. Discussion

In this work we set out to test the validity of earwax for measuring long-term glucose concentration by using a novel self-sampling outer-ear device. The main finding of the study is that by using the earwax self-sampling device, earwax was a more efficient specimen compared to HbA_1c_ in measuring glycemic levels. Furthermore, glucose measurements differently than HbA_1c_, were not affected by any of the covariates considered. Moreover, the novel device proved to be a feasible approach to rapidly extract wax for analysis with substantial time reduction compared to conventional methods.

We found that follow-up samples of glycemia, HbA_1c_, and EGC were significantly larger in comparison to their respective baseline concentrations. All associations between EGC and cross-sectional and longitudinal glycemic levels showed highly positive correlation coefficients. On the contrary, HbA_1c_ associations with the same short- and long-term glucose levels only exhibited low to moderate correlations. Earwax samples were up to 59% more predictable than HbA_1c_ specimens at reflecting the average glucose concentration over the preceding month period.

The earwax self-sampling device compared to the Reiner–Alexander syringe significantly reduced the time needed (7:43 h less) to analyze EGC. The novel device uses a dry method of extraction which bypassed the need to dry samples before analysis, a typical step of conventional water bases methods. The earwax self-sampling device processing time is comparable to HbA_1c_ methods, currently the gold-standard for measuring long-term glucose level without inconveniences and associated costs of bloodletting and significantly faster analysis of glycemic levels. Furthermore, while earwax glucose and HbA_1c_ analytic costs were similar, RT-CGM is significantly more expensive. The RT-CGM was lately developed to provide a glucose reading and trend levels collected every five minutes for up to seven days. Although RT-CGM may be a useful educational and motivational tool, diabetes self-management that includes the use of RT-CGM is likely to be more time-consuming for patients and force them to focus on different aspects of diabetes. Twice-daily self-monitoring of blood glucose is still required to calibrate the RT-CGM device and to inform treatment decisions in those using prandial insulin. Discrepancies between finger-stick blood glucose and sensor values may distress patients. Furthermore, high and low glucose threshold alarms may be disturbing. It has been reported that these devices produce a large amount of information that patients do not know how to handle it [45,46]. Nonetheless, HbA_1c_ and earwax glucose costs must be carefully interpreted since these were estimated in the UK and USA. Therefore, they might not accurately represent their current costs for other countries. Furthermore, the cost of the self-sampling earwax device and of the earwax glucose analysis considered were production costs. Production costs do not reflect the full commercial cost when the product becomes commercially available. In this context, HbA_1c_ might be the more affordable method for measuring chronic glucose concentrations. However, other advantages of this novel method, such as an increase accuracy in measuring chronic glycemia, the negligible risk of side effects and the practicality of its use, might overcome the potential higher cost. Hence, future clinicians, patients or both might prefer the self-sampling earwax device over HbA_1c_ as shown in previous work [42]. Nevertheless, the approach requires patients’ compliance and meeting the standards of health systems across the world. In situations like the current COVID-19 pandemic this novel approach might be preferable in view of the restrictions in mobility and social contact, which could result in a more efficient way to monitor chronic glycemic levels. Furthermore, in the context of COVID-19, in view of the increase in prevalence of mental health conditions [47], a device which reliably measures chronic glucose levels unaffected by stress factors might be particularly useful. Future, economic evaluation research study should be conducted to deeply investigate these hypotheses.

We found stronger associations than previous studies in the correlations between HbA_1c_ and fasting and postprandial glucose levels among the general population. Van’t Riet et al., (2010) found correlations of only 46% and 33% when fasting plasma glucose and 2 h post-load plasma glucose were correlated with HbA_1c_ in a large sample of controls [11]. It has been shown that HbA_1c_ usually shows increased associations with fasting (71%) and postprandial glucose levels (79%) among diabetic patients, rather than in controls [11]. This improved HbA_1c_ association with the postprandial glycemic level in diabetic patients is, however, smaller than the follow-up ECG/PSG correlation found here. On the other hand, the HbA_1c_ association with fasting glycemic level in diabetes was exactly the same that the one that we found here between baseline-EGC and FSG. It may be possible that EGC also shows an improved correlation in this metabolic disorder. Future studies may correlate PSG and FSG with EGC in diabetic patients.

Differences in the period covered by the baseline and the follow-up earwax sample may explain why the correlation between the baseline EGC and FSG (R = 0.71, *p* < 0.001) was much smaller than the association found between the follow-up EGC and PSG (R = 0.90, *p* < 0.001). This might be explained by the fact that only the amount of secreted earwax at the time of follow-up was standardized. Hence, the baseline period covered by the baseline EGC varied among participants and might have been affected by a range of factors. Aside from biological differences in fasting and postprandial glycemic levels, peaks of hyperglycemia, due to episodes of physical activity or stress before their inclusion in the study, might have affected baseline earwax measurements. This result suggests that EGC equally weights episodes of fasting and postprandial glucose levels. Conversely, HbA_1c_ is indeed greatly influenced by FSG than PSG. People spend more time fasting than eating during 24 h [9]. We also found an increased correlation between HbA_1c_ and FSG (R = 0.51) than HbA_1c_ with PSG (R = 0.47). The follow-up EGC may not be completely comparable with PSG. We recently showed that the earwax self-sampling device was significantly more efficient than the Reiner–Alexander syringe at removing earwax from healthy outer ears [48]. This suggests that some residual amount of earwax may have been left by the Reiner-Alexander method we used in this study that could have been extracted by the novel device. This would mean that the follow-up earwax sample extracted by the earwax self-sampling device may have also contained some residual earwax, and thus predominantly, but not exclusively, represented the EGC of the last month. The follow-up HbA_1c_ may also not be entirely comparable with PSG. HbA_1c_ is widely used as an index of the average level of glucose concentration over the preceding three months, although several studies have found that HbA_1c_ is predominantly influenced (75%) by the average concentration of glucose levels of the previous one month [12]. Therefore, both follow-up samples of EGC and HbA_1c_ predominately, but not exclusively represented the mean blood sugar over the last month. Future studies should investigate the same period of glucose concentration and correlate the mean of blood sugar with a follow-up ECC sample that is obtained after a baseline cleaning procedure that also used the earwax self-sampling device.

EGC was better than HbA_1c_ for reflecting acute levels of glycemia. All correlations between EGC and FSG and PSG were stronger than the observed coefficients when HbA_1c_ was associated with the same levels of glycemia. Furthermore, EGC showed the largest difference with HbA_1c_ correlations when EGC was associated with the mean blood sugar studied here. Indeed EGC/mean blood sugar correlation (R) was almost 50% stronger than the HbA_1c_ association with the same mean of glucose level. Furthermore, earwax was approximately 60% more accurate in predicting chronic glucose levels (R^2^) than HbA_1c_. This suggests that earwax is not only better than HbA_1c_ for reflecting acute glucose levels, but also for chronic. Future studies may correlate EGC with mean blood sugar over different periods.

In relation to confounders, we found, as previous studies have, that HbA_1c_ levels are affected by age [2]. We also found that HbA_1c_ was affected by the level of education. Participants’ type of employment may likely have explained this. It has been shown that jobs that require highly educated workers are also associated with increased working hours [49], which, in turn, are associated with increased HbA_1c_ [22]. Indeed, even though participants included in this study were healthy, they were exposed to a significant number and hassles and life events when compared with other healthy research samples originating from Chile [50] likely affecting their self-reporting of stressful events [40]. However, these events were most likely within the remit of stressful jobs or studies, considering that 43.2% of them were undergraduate students or had graduated from University. We also verified previous results that indicated that smoking decreased FSG and PSG. Earwax, however, was a more stable specimen since its glucose levels were not affected by any short- or long-term covariate studied here.

Two samples may appear to be a small number to reflect the average glucose concentration over one month, especially when considering that glucose is a substance with a variable profile of secretion. Indeed, some studies have used the area under the curve formula using several time points of glucose samples across the day for estimating the average concentration in this sugar [51]. The mean between fasting and postprandial glycemic levels, however, has proven to be a predictable index for reflecting the average concentration of glycemia. In fact, this index is also used with diabetic patients. Svendson and co-workers found that the average glucose level derived from approximately 2 to 300 blood measurements from 18 Type 1 diabetes patients correlated almost perfectly (R = 0.96) with HbA_1c_ [52]. Ozmen et al. found that the mean plasma glycemic level derived from fasting and postprandial plasma glucose levels also correlates strongly with HbA_1c_ in Type 2 diabetic patients [53]. Recently, the mean between postprandial and fasting glycemic levels was also used for monitor treatment in women with gestational diabetes mellitus [54]. The mean blood sugar between FSG and PSG may be even more valid and reliable for estimating the average glucose concentration among healthy people. Controls present less variability in their 24 hr glucose levels compared to diabetic patients [55]. Nonetheless, it may be more accurate to say that the estimated glucose mean of this study was obtained from longitudinal values, rather than the chronic glucose level.

A randomized study, blind to the intervention, may be another way to test the hypothesis that earwax glucose is more predictable than HbA_1c_ for measuring chronic glucose levels especially if several time points are considered. Inter-individual differences related to participants’ abilities to absorb different meal components may have also an effect on their glucose levels [56]. Some studies use the glucose tolerance test after the intake of 75 g of glucose, rather than postprandial levels after the intake of one standardized meal. We used Ensure^®^, as standardized meals contain glucose and several other nutrients, which may have different absorption rates, affecting, PSG levels. Moreover, the PSG test that we used has been widely used in several other research projects [57,58]. This is because, in comparison to normal meals, Ensure^®^ is easier to absorb due to its liquid characteristics. Furthermore, we excluded any participants with food allergies, such as lactose intolerance, which may have altered the absorption rates of some nutrients. With regards to the differences between plasma and serum, some studies report that plasma glucose is higher than serum glucose, whereas other studies found no difference [59]. The measurement of glucose in serum is not recommended for the diagnosis of diabetes [60]. We did not use FSG or PSG to make any diagnosis, we recruited a sample of healthy participants to investigate their glucose levels using different specimens.

## 5. Conclusions

Earwax showed to be more predictable than HbA_1c_ at reflecting acute and chronic glucose levels in healthy people. Earwax was also a more stable specimen since it was not affected by any confounders. Future larger validation longitudinal studies could correlate a higher number of fasting and postprandial plasma glucose samples with EGC and consider randomization to confirm the superiority of earwax methods. The earwax self-sampling device proved to be an effective method to measure EGC and may be utilized in diabetes and other metabolic disorders. EGC using the novel device may be a harmless, economic, and suitable test for measuring long-term glucose concentrations.

## Figures and Tables

**Figure 1 diagnostics-10-01069-f001:**
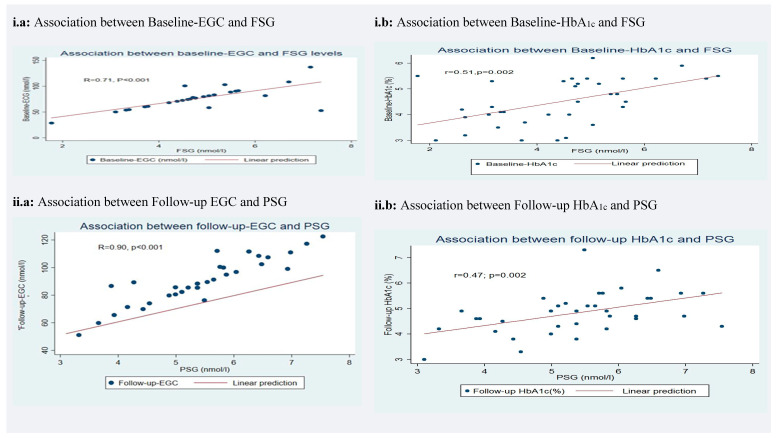
Figures (**a**) of the left side: EGC correlations with FSG and PSG samples at the baseline (**i**) and follow-up (**ii**) visits. Figures (**b**) of the right side: HbA_1c_ correlations with FSG and PSG samples at the baseline (**i**) and follow-up (**ii**) visits. Figures (**iii**): Chronic glycemia correlations with the baseline EGC (**a**) and HbA_1c_ (**b**) samples. Figures (**iv**): Chronic glycemia correlations with follow-up EGC (**a**) and HbA_1c_ (**b**) samples. The chronic glycemic level was calculated as the mean between FSG and PSG. The inserted value was a coefficient of Pearson correlation, glucose levels, and *p*-value.

**Table 1 diagnostics-10-01069-t001:** Socio Demographic Variables.

Variable		Results
**N Participants** **Females (N, %)**		37, (20, 54.1)
**Age (Years)** Mean (SD)		29.9, (1.4)
**Civil status: single (yes),** N (%)		32; (86.5)
**Undergraduate or graduates** N, (%)		16; (43.2)
**Ethnicity**	Mixed race, *n* (%)	36, (96.3)
	White *n* (%)	1, (3.7)
**Alcohol**	(yes) ^δ^, *n* (%)	10, (27.0)
	Units ^φ^ mean, (SD)	1.3; (0.5)
**Tobacco (yes),** *n* (%)		9, (24.3)
**Contraceptive pill (yes),** *n* (%)		9, (52.9)
**Medical or psychiatric comorbidity,** *n* (%)		0, (0)
**Medication,** *n* (%)		0, (0)

^δ^: at least one unit last week and: any medication, including psychotropic and steroidal medication. ^φ^: One alcohol unit is measured as 10 mL or 8 g of pure alcohol. This equals one 25 mL single measure of whisky (Alcohol by volume (ABV) 40%), or a third of a pint of beer (ABV 5–6%) or half a standard (175 mL) glass of red wine (ABV 12%).

**Table 2 diagnostics-10-01069-t002:** Anthropometric results.

Variable		Q1	Median	Mean, (SD)	Q3
**Height** (cm) Mean, (SD)	Whole sample	160	167	166.7, (1.4)	173
	Female	157	160	161.6, (1.8)	166
	Male	168	173	172.7, (1.3)	176
**Weight** (kg) Mean, (SD)	Whole sample	62	72	72.5, (2.5)	78
	Female	57.5	65.5	64.6, (2.0)	72
	Male	72	75	81.8, (3.9)	95
**BMI** (kg/m^2^), Mean, (SD)	Whole sample	23.3	24.9	25.6, (0.6)	26.7
	Female	22.8	24.6	24.2, (0.6)	25.5
	Male	24.1	25.4	27.2, (1.1)	31.2
**Waist circumference** (cm), Mean, (SD)	Whole sample	77	86	85.9, (2.4)	95
	Female	70.5	78	78.8, (2.3)	87
	Male	88	93	94.4, (3.4)	102

BMI: Body Mass Index.

**Table 3 diagnostics-10-01069-t003:** Self-Administrated Questionnaires.

Questionnaire	Results
Perceived Stress Scale (PSS), **Mean (SD)**	22.6, (1.1)
Life events score (RLCQ), **Mean (SD)**	141.2, (20.8)
History of severe life events (RLCQ) **(last month), N (%)**	10, (27.0)
Number of Hassles (last month), **Mean (SD)**	16.7, (1.7)
Severity index of hassles, **Mean (SD)**	22.9, (2.8)
Subjects under increased number (>25) of hassles **(last month),** **N (%)**	9; (24.3)
Subjects having problems dealing with their hassles **(last month),** **N (%)**	1, (2.7)

RLCQ; Recent Life Change Questionnaire, PSS: Perceived Stress Scale.

**Table 4 diagnostics-10-01069-t004:** Baseline and follow-up glycemia, HbA_1c_, EGC, and time comparisons.

Assessment	Baseline				Follow-up				*p*-Value ^ω^
Time in the morning, H ± SD	8:41 ± 00:50				10:53 ± 00:44				<0.05
**Sample**	**EGC** (nmol/L)				**EGC** (nmol/L)				
	Q1	Median	Mean (s.d)	Q3	Q1	Median	Mean (s.d)	Q3	
	60.5	76.9	76.7; (4.0)	82.5	81.5	88.9	94.7; (2.9)	101.5	<0.01 *
**Sample**	**HbA_1_c ^¥^** (%)				**HbA_1_c ^¥^** (%)				
	Q1	Median	Mean (s.d)	Q3	Q1	Median	Mean (s.d)	Q3	
	3.9	4.5	4.5; 0.9	5.4	4.3	4.9	4.8; 0.8	5.4	0.02 *
**Sample**	**FSG** (nmol/L)				**PSG** (nmol/L)				
	Q1	Median	Mean (s.d)	Q3	Q1	Median	Mean (s.d)	Q3	
	3.2	4.5	4.3; (0.3)	5.2	4.9	5.4	5.4; (0.2)	6.0	<0.01 *

^ω^: *p*-values were obtained using repeated t-test. *: *p*-value was significant at 0.05 level. HbA_1c_: Glycated hemoglobin. FSG: Fasting Serum Glucose. PSG: Postprandial Serum Glucose. ^¥^: HbA_1c_ was calculated using the National Glycohemoglobin Standardization Program (NGSP) unit which expresses the percentage of HbA_1c_ over the total amount of hemoglobin.

**Table 5 diagnostics-10-01069-t005:** Time required to analyse different specimens.

Procedure	Time Required (Hours)			
	EGC Using the Syringe	Glucose Using Earwax Self-Sampling Device	HbA_1c_ in Serum	Glucose in Serum
Centrifugation of the sample.	00:00	00:00	00:20	00:20
Pre-extraction drying of the sample with N_2_.	08:30	00:47	00:00	00:00
Extraction of the sample using an organic solvent.	02:10	02:10	00:00	00:00
Post-extraction drying of the sample with N_2_.	00:40	00:40	00:00	00:00
Protocol of analysis.	01:00	01:00	04:00	01:00
**Total time**	**12:20**	**04:37**	**04:20**	**01:20**

**Table 6 diagnostics-10-01069-t006:** Estimated costs related to different sampling methods for measuring chronic glucose over a three-month period in the UK.

Method	Associated Costs	Price (£)	
HbA_1c_	Healthcare staff ^Š^	17.04	
	Analysis ^§^	9.07	
	**Total**	**26.11**	
RT-CGM ^&^	Monitor without insulin pump	1000.00	n/a
	Monitor with insulin pump	n/a	500.00
	Sensors (3 months)	360.00	360.00
	**Total**	**1360.00**	**860.00**
Earwax Glucose *	Self-sampling device	8.07	
	Analysis	30.00	
	**Total**	**38.07**	

n/a: Not applicable. HbA_1c_: Glycated hemoglobin; ^&^RT-CGM: The Realtime Continuous Glucose Monitoring, and: Values were obtained from NHS UK at https://www.nhs.uk/conditions/type-1-diabetes/continuous-glucose-monitoring-cgms/. *: Novel self-sampling device production cost was provided by Trears Ltd. www.trears.com. ^Š^: The average cost per hour of a Staff nurse (Band 6), Specialist registrar (middle band), Junior doctor and Diabetes specialist nurse (Band 6) in the UK in 2017 [43]. ^§^: in Ko, S. Q., Quah, P. and Lahiri, M. The cost of repetitive laboratory testing for chronic disease. *Intern. Med. J.* 49, 1168–1170 (2019) [44].

**Table 7 diagnostics-10-01069-t007:** Different correlations of baseline and follow-up measures of EGC and HbA_1c_ with FSG, PSG, and mean glycemia.

	FSG (nmol/L)			PSG (nmol/L)			Mean Glycemia (nmol/L)		
	R	R^2^	*p*-Value	R	R^2^	*p*-Value	R	R^2^	*p*-Value
**Baseline-EGC** (nmol/L)	0.71	0.49	<0.001						
**Baseline-HbA_1c_** (%)	0.51	0.26	0.002						
**Follow-up-EGC** (nmol/L)				0.90	0.81	<0.001			
**Follow-up-EGC** (%)				0.47	0.22	0.002			
**Baseline-EGC** (nmol/L)							0.62	0.38	<0.001
**Baseline-HbA_1c_** (%)							0.55	0.30	<0.001
**Follow-up-EGC** (nmol/L)							0.84	0.71	<0.001
**Follow-up-HbA_1c_** (%)							0.35	0.12	0.03

R: Regression Coeffient. R^2^:Coeffient of Determination. FSG: Fasting Serum Glucose. PSG: Postprandial Serum Glucose. The average glucose level was calculated as the mean between FSG and PSG. HbA_1c_ was calculated using the National Glycohemoglobin Standardization Program (NGSP) units which expresses the percentage of HbA_1c_ over the total amount of hemoglobin.

**Table 8 diagnostics-10-01069-t008:** Linear regression models between several covariates and baseline and follow-up samples of earwax, glycemic and HbA_1c_.

Variables	Baseline- EGC (nmol/L)			Follow-up- EGC (nmol/L)			Baseline- HbA_1_c * (%)			Follow-up- HbA_1_c (%)			FSG (nmol/L)			PSG (nmol/L)		
	β	*p*-Value	CI	β	*p*-Value		β	*p*-Value	CI	β	*p*-Value	CI	β	*p*-Value	CI	β	*p*-Value	CI
Age	0.4	0.38	−0.6; 1.4	0.3	0.29	−0.3; 1.0	0.3	0.14	<−0.1; 0.1	<0.01	<0.01 *	<0.1; <0.1	<0.01	0.27	<−0.1; <0.1	<0.1	0.07	<−0.1; <0.1
Sex	−5.2	0.50	−22.3; 11.3	2.1	0.70	−9.3; 13.5	−0.2	0.43	−0.9; 0.4	<−0.1	0.86	−20.5; 16.0	−0.1	0.84	−1.0; 0.9	0.1	0.81	−0.6; 0.8
Graduates or undergraduate ^ξ^	5.1	0.54	−11.9; 22.1	9.1	0.10	−1.9; 20.0	0.4	0.22	−0.3; 1.1	0.7	0.01 *	0.2; 2.0	0.4	0.35	−0.5; 1.4	0.7	0.04 *	<0.1; 1.4
Alcohol (unit) ^φ^	4.0	0.18	−1.9; 10.0	−1.3	0.19	−3.4; 0.7	<0.1	0.73	−0.1; 0.1	<−0.1	0.70	−0.1; 0.1	−0.1	0.51	−0.2; 0.1	−0.1	0.18	−0.2; <0.1
Tobacco	−9.9	0.40	−33.7; 13.9	−10.5	0.13	−23.2; 3.1	−0.5	0.19	−1.2; 0.3	−0.4	0.22	−1.0; 0.2	−1.3	0.01 *	−2.3; −0.3	−0.9	0.02 *	−1.7; −0.1
BMI (kg/cm^2^)	−1.3	0.17	−3.3; 0.6	−0.9	0.18	−2.4; 0.5	<0.1	0.97	−0.1; 0.1	0.1	0.61	−0.1; 0.1	<−0.1	0.74	−0.1; 0.1	<−0.1	0.43	−0.1; 0.1.
Waist circumference (cm)	−0.24	0.36	−0.8; 0.3	−0.1	0.72	−0.5; 0.3	<−0.1	0.42	<−0.1; <0.1	<−0.1	0.36	<−0.1; <0.1	<−0.1	0.73	<−0.1; <0.1	<−0.1	0.68	<−0.1; <0.1
Contraceptive pill	10.2	0.17	−5.0; 25.4	4.2	0.53	−9.7; 18.0	−0.2	0.70	−1.2; 0.8	−0.1	0.86	−0.8; 0.7	0.74	0.13	−0.2; 1.7	0.1	0.74	−0.6; 0.8
PSS	−0.8	0.21	−2.1; 0.5	<−0.1	0.94	−0.9; 0.8	<−0.1	0.93	<−0.1; <0.1	<0.1	0.23	<−0.1; 0.1	<0.1	0.88	−0.1; 0.1	<0.1	0.57	<−0.1; 0.1
Number of Hassles	<−0.1	0.94	−0.9; 0.9	−0.3	0.23	−0.8; 0.2	<0.1	0.49	<−0.1; <0.1	<0.1	0.36	<−0.1; <0.1	<0.1	0.84	<−0.1; <0.1	<−0.1	0.66	<−0.1; <0.1
Severity of Hassles	<0.1	0.98	−0.5; 0.5	−0.1	0.45	−0.4; 0.2	<0.1	0.41	<−0.1; <0.1	<0.1	0.15	<−0.1; <0.1	<0.1	0.78	<0.1; <0.1	<−0.1	0.92	<−0.1; <0.1
RLCQ	<−0.1	0.54	<−0.1; <0.1	<0.1	0.79	<−0.1; <0.1	<−0.1	0.72	<−0.1; <0.1	<−0.1	0.09	<−0.1; <0.1	<−0.1	0.83	<−0.1; <0.1	<−0.1	0.97	<−0.1; <0.1
Severe RLCQ	3.0	0.54	−13.3; 7.2	<−0.1	0.99	−7.0; 7.0	<0.1	0.81	−0.3; 0.4	−0.2	0.20	−0.5; 0.1	−0.1	0.86	−0.6; 0.5	<−0.1	0.88	−0.5; 0.4

*: HbA_1c_ was calculated using the National Glycohemoglobin Standardization Program (NGSP) units which expresses the percentage of HbA_1c_ over the total amount of hemoglobin. PSS: Perceived Stress Scale. RLCQ: Recent Life Event Questionnaire. HbA_1c_: Glycated hemoglobin. EGC: Earwax Glucose Concentration. FSG: Fasting Glycemic Levels. PSG: Postprandial Serum Glucose. ^φ^: One alcohol unit is measured as 10 mL or 8 g of pure alcohol. This equals one 25 mL single measure of whisky (Alcohol by volume (ABV) 40%), or a third of a pint of beer (ABV 5–6%) or half a standard (175 mL) glass of red wine (ABV 12%). ^ξ^: In comparison to those who were in their secondary studies or doing a technical work.

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
