# Peer review of "A Novel Earwax Method to Measure Acute and Chronic Glucose Levels"

_diagnostics, 2020, doi:10.3390/diagnostics10121069_

Round 1

Reviewer 1 Report

I would suggest adding Fig 1 to the body of the manuscript. It helps validate your measurement of ear wax glucose and serum glucose.

Author Response

Reviewer #1

Methods: ΟΚ

Background & References

It can be improved

Thank for this. We have completely reviewed the manuscript, introducing a new variable into analyses, such as the costs related to the novel method of sampling and its analysis, as requested by the other reviewer (please see page 2 highlighted in yellow). We have also discussed the poor reliability of HbA1c for detecting episodes of hypoglycaemia (page 2). Ultimately, we have also introduced our most recent results that strongly suggests that earwax might also be able to cumulate other substances, such as cortisol (page 3). All these points have been supported with their respective references. We hope that these points have improved the quality of our work.

Research Design

It can be improved

Thanks for this. Unfortunately, there is not too much that we can do to amend the study design chosen by us since the study has already been completed. Nonetheless, we have discussed its limitations, as the fact that it might be more accurately to say that we associated earwax glucose and HbA1c with longitudinal, rather than chronic glycemia (page 16). Therefore, future studies should include an increased number of blood sugar samples that covers the time frame covered by the study. However, it is also important to bear in mind that Svendson and co-workers have already averaged glucose level derived from approximately 2 to 300 blood measurements from 18 Type 1 diabetes patients correlated almost perfectly (R=0.96) with HbA1c 1. Therefore, it seems unlikely that a larger number of glycaemic samples in healthy people might significantly vary our results.

We also suggest that a Randomised Clinical Trial (RCT), rather than a prospective case-control study might be a better study design to test our hypotheses (page 16). However, acknowledging that there is no perfect study we believe that our strong results do support the study design chosen by us.

Methods

It can be improved

Thanks for this. We had forgotten to include the statistical analysis that we used to graph the regression line. Then, this has now been included in the methods (please see page 4).

Results

It can be improved

Thanks for this. We have now considered the results relate to our new analysis that includes the costs related to the novel method of sampling and analysis. However, we believe that they need to be carefully interpretated since our study was not primarily designed to test this hypothesis. Future, economic evaluation research study should be conducted to address these questions (page 13).

Conclusions

It can be improved

Thanks for this. We found that our conclusion did not experience a significant change even though the inclusion of all your accurate comments.

Figures

I would suggest adding Fig 1 to the body of the manuscript. It helps validate your measurement of ear wax glucose and serum glucose.

Thank for this. Yes, we also agreed with you here. We agree with the fact that it was a mistake including that figure within the supplementary section. That is why it is now part of the main body, as suggested (pages 9 to 11).

References:

(1)       Aaby Svendsen, P.; Lauritzen, T.; S�egaard, U.; Nerup, J. Glycosylated Haemoglobin and Steady-State Mean Blood Glucose Concentration in Type 1 (Insulin-Dependent) Diabetes. Diabetologia 1982, 23 (5). https://doi.org/10.1007/BF00260951.

Reviewer 2 Report

“The aim of this study was to validate the use of earwax for measuring long-term glucose concentration by using a novel, self-cleaning outer ear device which does not require any technical expertise… EGC was 59% more accurate in predicting glucose levels than HbA1c for measuring longitudinal (chronic) glucose concentration. Earwax samples were not affected by any of the covariates considered… However future studies may correlate PSG and FSG with EGC in diabetic patients”.

Several times the Authors stress the lower cost of this method: it is useful to insert a table with the various costs (in analogy to supplementary table 4).

Insert supplementary table 6 in the text.

Insert the number relating to references 36 and 37 in the text.

It would also be important to evaluate the method's ability to document any hypoglycemic levels.

Standardize the References section according to international standards: uppercase / lowercase (references 9, 30, 50); start / end page (eg references 3, 9, 10,12,17… 42, 48); incompleteness of references 31, 35, 50, 51.

Author Response

Reviewer #2

Results

  • It would also be important to evaluate the method's ability to document any hypoglycemic levels.

Thank for this. This is a really good point that we had not considered before. Much appreciated. In fact, the poor reliability of HbA1c for detecting episodes of hypoglycaemia reinforces the need of introducing a more reliable specimen, as earwax might be for reflecting chronic glycaemia (please see page 2 highlighted in yellow).

Figures

  • Several times the Authors stress the lower cost of this method: it is useful to insert a table with the various costs (in analogy to supplementary table 4).

Thank for this. This is also another good point. Much appreciated. Therefore, we have now discussed our findings related to the costs related to the novel method of sampling and analysis (page 13). However, we believe that they need to be carefully interpretated since our study was not primarily designed to test this hypothesis. Future, economic evaluation research study should be conducted to address these questions.

  • Insert supplementary table 6 in the text.

Thanks for this. This table has now been included in the main body of the paper (please see page 12).

References

  • Insert the number relating to references 36 and 37 in the text.

Our apologies. This has now been amended (please see page 3)

  • Standardize the References section according to international standards: uppercase / lowercase (references 9, 30, 50); start / end page (eg references 3, 9, 10,12,17… 42, 48); incompleteness of references 31, 35, 50, 51.

Sorry about this. All references have now been double-checked and their typos amended.